# Spontaneous discovery of novel task solutions in children

**Nicolas W. Schuck**[1,2]*, **Amy X. Li**[3], **Dorit Wenke**[4], **Destina S. Ay-Bryson**[5], **Anika T. Loewe**[1,2], **Robert Gaschler**[6], **Yee Lee Shing**[7,8]

**1** Max Planck Research Group NeuroCode, Max Planck Institute for Human Development, Berlin, Germany, **2** Max Planck UCL Centre for Computational Psychiatry and Ageing Research, Berlin, Germany, **3** School of Psychology, University of New South Wales, Sydney, Australia, **4** PFH Private University of Applied Sciences, Göttingen, Germany, **5** Department of Rehabilitation Sciences, Humboldt University of Berlin, Berlin, Germany, **6** School of Psychology, FernUniversität Hagen, Hagen, Germany, **7** Institute of Psychology, Goethe University Frankfurt, Frankfurt, Germany, **8** Center for Lifespan Psychology, Max Planck Institute for Human Development, Berlin, Germany

\* schuck@mpib-berlin.mpg.de

**Data Availability Statement:** Anonymized data and code used to generate all results are publicly available on github https://github.com/nschuck/Schucketal_PLoSONE_2022, DOI: 10.5281/zenodo.6481202.

## Abstract

Children often perform worse than adults on tasks that require focused attention. While this is commonly regarded as a sign of incomplete cognitive development, a broader attentional focus could also endow children with the ability to find novel solutions to a given task. To test this idea, we investigated children's ability to discover and use novel aspects of the environment that allowed them to improve their decision-making strategy. Participants were given a simple choice task in which the possibility of strategy improvement was neither mentioned by instructions nor encouraged by explicit error feedback. Among 47 children (8—10 years of age) who were instructed to perform the choice task across two experiments, 27.5% showed a full strategy change. This closely matched the proportion of adults who had the same insight (28.2% of n = 39). The amount of erroneous choices, working memory capacity and inhibitory control, in contrast, indicated substantial disadvantages of children in task execution and cognitive control. A task difficulty manipulation did not affect the results. The stark contrast between age-differences in different aspects of cognitive performance might offer a unique opportunity for educators in fostering learning in children.

## Introduction

Humans develop into remarkably adaptive and efficient decision makers over the first two decades of their lives. Of particular importance for this process is the development of cognitive control functions, which allow us to keep information about the ongoing task in working memory and shield it from interference by irrelevant distractions [1, 2]. Developmental research has therefore often focused on children's improvements in these functions [3–7].

Yet, truly *flexible* goal-directed behavior also involves improving one's current decision making strategy. Similar to how discovering unknown connections can allow shortcuts in navigation, learning about previously ignored or novel contingencies in the environment can lead

**Funding:** NWS was funded by an Independent Max Planck Research Group grant awarded by the Max Planck Society (M.TN.A.BILD0004, www.mpg.de) and a Starting Grant from the European Union (ERC-StG-REPLAY-852669, www.erc.europa.eu/). DW was funded by DFG grant (WE2852/3-1, www.dfg.de). YLS was funded by a Minerva Research Group by the Max Planck Society (www.mpg.de), a Starting Grant from the European Union (ERC-StG-PIVOTAL-758898, www.erc.europa.eu), and a Fellowship from the Jacobs Foundation (JRF 2018-2020, www.jacobsfoundation.org). AL is supported by the International Max Planck Research School on Computational Methods in Psychiatry and Ageing Research (IMPRS COMPPPSYCH, www.mps.ucl-centre.mpg.de). The funders had no role in study design, data collection and analysis, decision to publish, or preparation of the manuscript.

**Competing interests:** The authors have declared that no competing interests exist.

to behavioral or cognitive changes that achieve the same goal in a more efficient manner. The disadvantage that children have with executing some tasks could hence lead to a somewhat paradoxical advantage for finding better strategies; "shortcuts" which can only be discovered by processing information that is irrelevant for the current strategy. In line with this idea, children have been found to outperform adults in detecting changes in shapes they were not cued to attend and in remembering information that is irrelevant for the instructed task [8]. This suggests that children may tend to distribute attention across multiple aspects of stimuli, including those that are not relevant to the instructed goal, and might also reflect children's comparatively high sensitivity to statistical regularities in their environment [9, 10]. Other research has also suggested that children may be more eager to explore less known options than adults [11], or to sample hypotheses in a more probabilistic fashion [12]. Likewise, a number of previous findings have shown that children are remarkably variable in the strategies they employ, when even performing the same task [13, 14] and emphasized that children usually use a variety of approaches to problem solve [15–17]. Frequent task switching, in turn, is known to weaken task maintenance or 'shielding' [18]. In combination, these characteristics might allow children to be surprisingly good in adaptive strategy updating, although it is regarded as a complex computational problem [19, 20].

This idea stands in contrast to a large developmental literature that has shown that efficient decision making is comparatively slow to develop [7, 21]. Compared to the development of other cognitive faculties, such as language or motor skills, decision making that involves multiple features becomes mature particularly late in development, and reaches adult-levels only in late adolescence [22, 23]. Likewise, the ability to focus attention on task-relevant aspects and to suppress distracting information has been found to be less effective in children in a variety of tasks, such as the anti-saccade [24, 25], Flanker [26] or Stroop [27] tasks and working memory capacity also does not reach adult levels until late adolescence [28]. Interestingly, even the ability to follow explicit rules continues to enhance as children become older in middle childhood, thereby contributing to the protracted development of children's control of behavior [4]. Over the same period of time, children become increasingly able to integrate and execute different rules according to the cues provided by task context [5], particularly starting from late childhood on [22]. Finally, model-based decision making is also known to develop slowly [29]. Neuroscientific research has linked the protracted cognitive development to the relatively delayed maturation of the prefrontal cortex, e.g. [30, 31].

The research summarized above suggests that cognitive control skills, and their underlying neural processes, undergo protracted development. Considerably less is known, however, to what extent the development of cognitive control is related to children's ability to flexibly update decision-making strategies. The main goal of the present paper is therefore to ask how good children are in discovering and updating an ongoing decision-making process with an alternative strategy that achieves the same goal. As we noted above, the combination of high sensitivity to statistical regularities with a lower ability to inhibit irrelevant information and to follow instructions, may lead children to be surprisingly good in discovering novel solutions to an instructed task. The second goal of this study is to examine to what extent alternative strategy discovery is related to cognitive control, particularly inhibition and working memory, as measured with independent covariate tasks. We hypothesized a negative relationship between inhibition and alternative strategy discovery, particularly in children.

## Experiment 1

Experiment 1 tested children and young adults with the Spontaneous Strategy Switch Task, which assesses the ability to discover and implement a novel strategy [32, 33]. Participants

were instructed to perform a simple decision making task that required responding to the spatial location of a stimulus (four possible locations) with two different buttons. Unbeknownst to participants, the stimulus color (two possible colors) was fully correlated with the required response, such that participants in principle could use an alternative, simpler strategy and respond to stimulus color (2 to 2 mapping) rather than stimulus location (4 to 2 mapping). The above mentioned previous work with the same task has shown that about one third of adult participants will discover and use the alternative strategy. The same data also indicated that strategy switches occurred abruptly (within a few minutes) and occurred throughout the experiment. In Experiment 1 we asked how frequent strategy discovery is among children compared to adults, and if the characteristics of strategy change differ between age groups.

## Materials and methods

**Participants.** Twenty-eight children and 22 young adults were tested in Experiment 1. Participants were excluded if they failed to perform the instructed *color* task (see below), as tested by a binomial test assessing performance against chance in the final two blocks (blocks 9 and 10, $\alpha$ = 5%). This led to the exclusion of 6 children and 1 adult. The effective sample size therefore consisted of 22 children (11 female) with a mean age of 9.5 years (SD = 2.5, range = 8 to 10 years) and 21 young adults (8 female) with a mean age of 22.7 years (SD = 0.8, range = 20 to 30 years). All participants and in the case of children their legal guardians provided informed consent and all applicable ethical regulations related to research with human participants were followed. The ethics board of the Max Planck Institute for Human Development approved all reported studies (project title: Erkennung von Punktewolken (PU2D)).

**Main task.**   *Stimuli*. Each stimulus consisted of 72 small colored squares that were distributed uniformly over a rectangular patch (120 × 120 px), covering half of the patch area. The patch of colored squares was displayed within a grey reference frame that was slightly larger than the patch (150 × 150 px). The patch itself was presented centrally on the screen, but on each trial the reference frame was offset from the center by ± 5 px on the horizontal and vertical axes (see Fig 1A). The patch was therefore closer to one of the four corners of the reference frame. Offsets changed trial-wise and participants were instructed to decide *where* the patch was positioned within the frame, i.e to which of the four corners of the reference frame it was closest to. To respond, participants used two response keys ([x] and [,] marked with a white label on a QWERTZ keyboard). One key had to be pressed whenever the stimulus was closer to the upper left or the lower right corner of the frame, whereas the other key was correct for the opposed corners (lower left and upper right). The response to corner mapping was randomized across participants and shown to participants throughout the task.

On each trial, the squares that made up the patch had the same color and were either green or red. Participants were not informed that the colors had any meaning for the task and the colors indeed changed randomly during the first block (50% red and 50% green patches for each response button). Beginning with the second block, however, the stimulus color was consistently paired with the required response button (Fig 1B). This meant that in trials requiring a left response (upper left or lower right corner), the patch was for instance always green, whereas in trials requiring a right response the patch was always red. If this was noticed by a participant, it allowed her to change her decision-making strategy from selecting buttons based on patch location to responding based on patch color. The color-button mapping was counterbalanced across participants.

*Trial types*. The main task included four different trial types that involved slightly different response requirements (Fig 1C). In *standard* trials, the patch and the reference frame appeared simultaneously for 400 ms on the screen and participants could respond as instructed

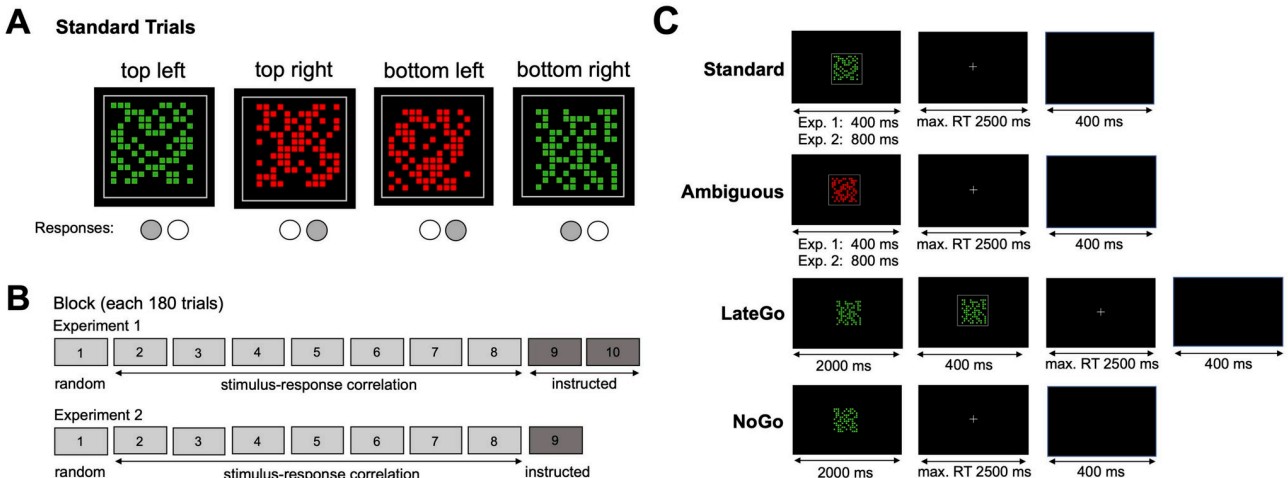

**Fig 1. Stimulus and task design.** (A): Stimulus response mapping in *standard* trials. The mapping was counterbalanced across participants. Each trial involved one patch of colored squares inside a light reference frame as shown. The colored squares were shifted systematically from the center of the frame and participants had to decide which corner of the grey frame the patch is closer to. (B): Block order for Experiments 1 and 2. Each block started with a block in which stimulus color and corner were uncorrelated ("random blocks"). Without notifying participants, from block 2 on the required response and the stimulus color had a fixed relation in all standard trials. After block 8, participants were instructed to use the color to determine their response ("instructed blocks"). Experiments 1 and 2 differed regarding the number of instructed blocks. (C): Trial structure for *standard*, *ambiguous*, *LateGo* and *NoGo* trials. Each row shows the onset and duration of the colored squares, the grey frame, the fixation cross and the response stimulus interval for one condition, see labels.

immediately after stimulus onset. In *LateGo* trials, the patch appeared for 2000 ms before the reference frame appeared for 400 ms in addition to the patch. Participants were instructed to withhold responding until the frame was displayed. *NoGo* trials were identical to LateGo trials, except that the frame did not appear after 2000 ms and the task continued with the next trial. Participants needed to withhold responding in these trials. *LateGo* and *NoGo* trials were included to measure inhibition. Finally, on *ambiguous* trials, patch and frame appeared simultaneously, but the frame was not offset from the center. Hence the patch was not closer to any of the four corners and responding based on relative spatial position of the patch would lead to random choice behavior. Responding based on color, however, would lead to choices in line with the stimuli's colors. These trials were therefore used to asses if and when participant began to use a color based strategy.

**Color task (instructed block).** Prior to block 9, participants were informed that stimulus color and the correct response were paired. As participants were not informed about the exact nature of the pairing, they had to find the relation themselves. The instructions stated that once the pairing was found, participants should base their responses on the color for the remainder of the experiment. Otherwise, the color task was identical to the main task in all regards. Performance in the color task was used to test if participants were able in principle to execute the strategy change of they had discovered it.

**Additional covariate measures.** *Questionnaire*. Following the main experiment, participants were asked to fill out a questionnaire containing several questions about the task. These questions asked (1) whether the hidden color rule was noticed [yes/no], (1b) if yes, when within the experiment it was noticed [participants indicated the proportion of elapsed time before noticing on a clockface], (2) whether the discovered color rule was used to make decisions [yes/no], (3) to report the rule by writing down which color was associated with which corner. Due to human error, questionnaire data from one adult participant were lost. Analyses

which considered the questionnaire data were constrained to include participants for which task and questionnaire data was available.

*Working memory test*. Participants completed a digit-sorting task as a measure of working memory (WM). For each trial, a set of numbers was verbally read out by the experimenter. After the last number was presented, participants were asked to write down the numbers in the ascending order on the answer sheet. A total of 15 sets of numbers divided into five levels were used, starting from four numbers at the first level and one number was added for each consecutive level. A set of numbers was assessed as incorrect if a number was missing or if the sequence was not in the correct order. A maximum of fifteen points could be scored on the task. Due to technical errors, WM data from 6 participants were lost (3 younger adults). Only analyses which considered the WM performance were constrained to include participants for which task and WM data was available.

*Stroop test*. A Stroop task was used as a measure of inhibition. The task consisted of 40 congruent, 40 incongruent, and 40 neutral trials. Participants were instructed to respond according to the font color of the stimulus word (e.g., for words shown in blue color, press the blue key). For congruent trials, the stimulus words ("BLUE" or "YELLOW") in their corresponding colors were presented on the screen. For incongruent trials, the stimulus words were shown with non-corresponding colors. For neutral trials, the stimulus word was "XXX" and was either shown in blue or yellow color. We computed two scores: the difference between reaction times in neutral and in congruent trials (semantic facilitation), and the difference between neutral and incongruent trials, the so called semantic interference score. In addition to data from participants whose WM scores were lost, Stroop data from two additional participants were lost. Analyses which considered Stroop performance therefore excluded two additional participants, relative to WM analyses.

**Procedures.**  The experiment began with instructions for the main task. Participants were explained that, on each trial, they should make a response based on the spatial position of the patch within the reference frame. While children received instructions verbally to ensure correct understanding, young adults read the same instructions themselves on the screen. Instructions did neither facilitate nor discourage color use, mentioning only that "each patch will be either red or green". Examples for each corner were shown in both colors. A printout showing the corner-response mapping was attached to the wall in front of the participants, allowing them to refer to it throughout the experiment (to reduce variance due to forgetting the mapping). Instructions explained all trial types and were followed by a pretraining that ensured that the rules were understood. The pretraining phase lasted for at least 50 trials and was ended once the participant made less than 20% errors in 24 consecutive trials. Participants received trialwise error feedback on the monitor during this part of the experiment, informing them when the given answer was incorrect, too late, or premature. Colors changed randomly in this part of the task.

After pretraining the main task started and lasted for 10 blocks of 180 trials each in Experiment 1 (Fig 1B). Each block contained 80 *standard*, 32 *ambiguous*, 32 *NoGo*, 16 *LateGo* trials. In order to discourage counting strategies, 12 additional trials were distributed randomly across trial types such that number of trials per condition varied slightly between blocks. Participants could take a short break after each block. During the main task, no trialwise feedback was given. If the block-wise error rate exceeded 20%, a warning about too many errors was displayed in the break between blocks.

During the first block ("random block"), the color in left and right response trials was chosen at random. This block was included to allow participants to settle into the task to a similar level. From Block 2 on, the color was associated with the correct response as described above. In the break before Block 9, participants were informed that the color and the response were

paired. They were not informed about the exact nature of the pairing but rather asked to find the relation and base their responses on the color for the remainder of the experiment ("instructed blocks"). This block was to examine whether participants could in principle discover the alternative color strategy given a strong hint. During this break, but before receiving instructions about the color task, participants were also asked to complete a questionnaire assessing knowledge of the color strategy (see above). Then they completed blocks 9 and 10. A subgroup of participants in Experiment 1 erroneously performed an additional 9th block before performing two instructed blocks (4 children and 5 young adults). This did not affect the first 8 blocks for those participants and data from this additional block were therefore not analyzed. After the main task and questionnaire were completed, participants performed the Stroop and working memory tasks. The overall duration of the experiment was approximately 160 minutes for children and 120 minutes for young adults.

**Analyses.** All analyses were performed using R [34], employing the 'lme4' package for mixed effects modelling [35]. Post-hoc tests were adjusted using the Tukey method as implemented in the package 'emmeans'. T-tests were corrected for variance inhomogeneity using the Welch test implemented in R. Unless otherwise noted, mixed effects models included a random intercept and slope of the linear factor Block per subject as well as fixed effects for the factors Block and Age group ('Young Adults' vs. 'Children'). To determine whether participants understood the task, we tested individually whether the percentage of correct regular trials in the color task was significantly different from chance (based on binomial test against chance at $\alpha$ = .05). This resulted in cut-offs of min. 65% correct color-based responses, ensuring that only performance of participants was analyzed who had the ability to perform the spatial task in principle.

*Switch point analysis.* We used the CUSUM method to determine the block when participants started using the color, as in [32]. For each participant, we first calculated the average percent of color use over all blocks. We then subtracted this overall mean from each block-wise mean, and calculated the cumulative sum of these differences. Because the differences are negative while the block-wise performance is below the overall mean, and positive once the percent color use is above the mean, the cumulative sum of the differences will decrease until a participant switches and start to increase afterwards. Switch time-points were therefore determined as the time-point at which each participants' cumulative difference score was at its minimum.

## Results

**Performance on main task: Standard trials.** Errors in blocks 1-8 during standard trials decreased with practice and consistently differed between children and young adults, as reflected in main effects of Block $\chi^2(1)$ = 8.6, $p$ <.001 and Age group, $\chi^2(1)$ = 32.3, $p$ <.001, respectively (see Fig 2A). Post hoc tests confirmed that the main effect of Age group was driven by younger adults committing less errors than children (25.3% vs. 7.7%, $p$ <.001). This difference persisted throughout the task and remained present in the last two blocks before the color instruction (blocks 7-8), $p$ <.001. No interaction between Age group and Block was found. Likewise, reaction times (RTs) differed between age groups, (988ms vs 653ms, $\chi^2(1)$ = 38.3, $p$ <.001) and decreased with practice, $\chi^2(1)$ = 41.5, $p$s <.001 (Fig 2B). Group differences persisted until the last blocks as evidenced by planned comparisons of the average RT in blocks 7 and 8, $p$ <.001.

Investigating performance during the final instructed block revealed that adults still outperformed children after participants had been provided with instructions to use color: error rates of children and adults were 10.5% vs. 2.6%, respectively (t-test: $p$ <.001). In addition, children

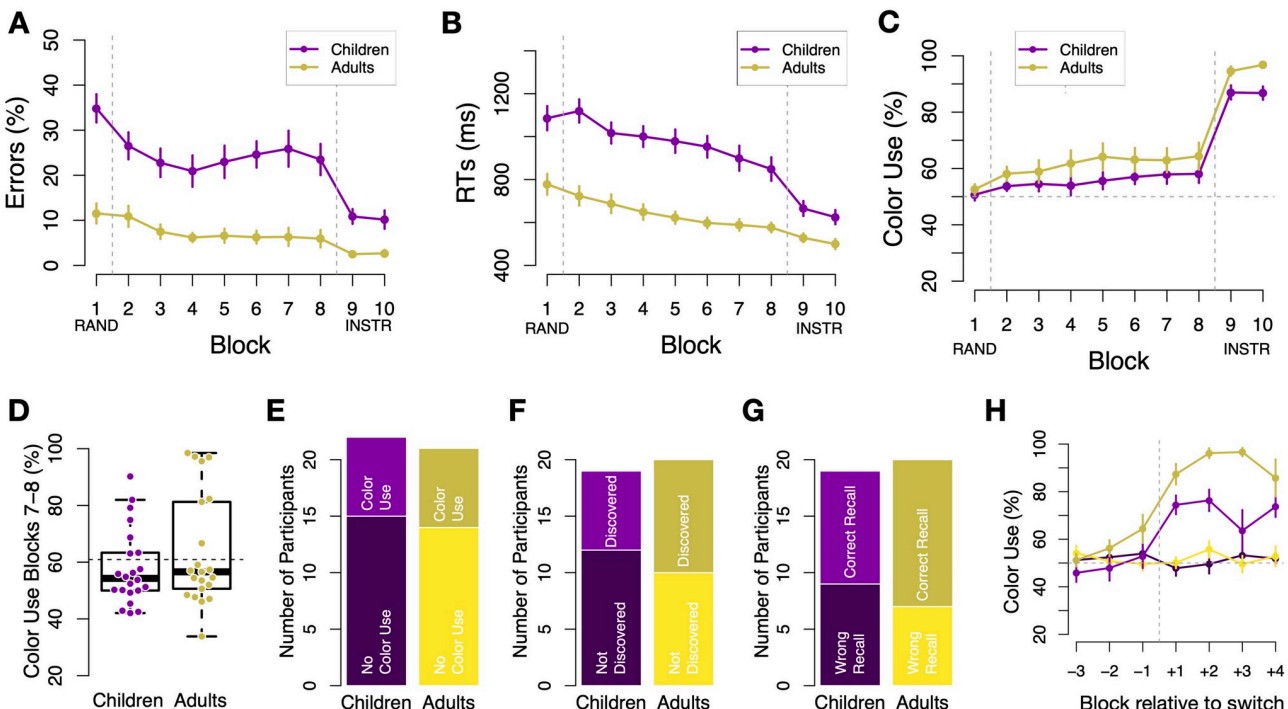

**Fig 2. Performance in standard trials and alternative strategy discovery.** A: Error rates as a function of block separately for children (purple) and younger adults (yellow) in Experiment 1. Large age differences in error rates persisted throughout all blocks of the main task. Children benefited more from color instructions provided in blocks 9-10. B: Average reaction times (RTs, in milliseconds) over blocks, also indicating sizable and persistent differences between children and young adults. Colors as in panel A. C: Percentage of color-based choices ("Color Use") in *ambiguous* trials as a function of block in both age groups (colors as as in A). No significant differences were found. D: Percentage of color use in Blocks 7 and 8, before instructions were given. Each dot reflects one participant. E: Proportion of participants whose behavior indicated a strategy switch towards color based responding by blocks 7 and 8 (>60% color use). No difference was found between age groups in this measure. F: Percentage of participants who reported discovery of the relation between colors and corners. No age group difference. G: Percentage of participants able to correctly report the color-corner association after block 8 (before color instructions were given). H: Percentage of color use in ambiguous trials time-locked to the mini-block in which a strategy switch was detected. To increase temporal resolution, blocks on the x-axis are split in half relative to C. Different participant numbers in G/H vs F reflect loss of questionnaire data. Bars represent s.e.m.

benefited more from the instructions than adults in terms of error rates, as evidenced by an interaction between Block (7/8 vs. 9/10) and Age group, $\chi^2(1) = 8.5$, $p < .001$. The same pattern was found concerning RTs, i.e. we found a main effect of Age group in Blocks 9/10 and an interaction between Block (7/8 vs 9/10) and Age group, $\chi^2(1) > 10$, $p$s $< .001$. Fig 2A and 2B shows participants performance data.

**Spontaneous strategy discovery and switch.** We next investigated participants' ability to discover and use the alternative strategy. We first assessed to what extent responses in ambiguous trials were based on stimulus color. For instance, if green was paired with left responses in standard trials, we measured the proportion of left responses in spatially ambiguous green trials and vice versa. A mixed effects model revealed an increase in color-based responding over time, i.e. a main effect of Block, $\chi^2(1) = 4.05$, $p = .04$, see Fig 2C. There was no evidence that children and adults differed in the extent of color use, i.e. no main effect of age group was found, $\chi^2(1) = 2.6$, $p = .10$. Pairwise t-tests showed no group differences during any of the blocks. Crucially, testing only behavior in the last 2 blocks before color instructions (i.e., mean of blocks 7 and 8), showed no difference between age groups, with average proportion of color based responding at 58.0% vs 63.7% in children and young adults, respectively, $\chi^2(1) = 1.28$, $p = .26$, see Fig 2D. Moreover, the proportion of participants who significantly used color

(binomial test $p <.05$) in the last two correlated blocks was 31.8% among children (7/22), 33.3% among young adults (7/21) and not statistically different between age groups, $\chi^2(1) = 0.01$, $p = 1$, see Fig 2E. This result was not affected by the choice of threshold (both $p$s >.28 when a higher threshold of at least 75% or a lower threshold of at least 50% color use were employed). Likewise, neither did the proportion of participants who verbally reported that they had discovered the color strategy differ between age groups ($p >.05$, Fig 2F), nor did the number of participants who could accurately report the association between color and corners in a questionnaire ($p >.05$, Fig 2G).

We next inferred the time point of discovery from participants' data (see Methods). Since no meaningful switch time could be inferred for participants who did not discover the novel strategy, their assigned time points were sampled randomly from the same distribution observed in switching participants. We then time-locked each participant's time course of color use to her/his inferred switch point. Because the distribution of switch time-points limits the number of trials before and after the switch for which data from all participants is available, we considered a time-window ranging from 3 half-blocks before to 4 half-blocks after the switch. As can be seen from Fig 2H, strategy discovery was abrupt, as in [32, 33, 36]. Pairwise tests in young adults between adjacent blocks showed no evidence of change in color use before or after the switch, while the switch itself was characterized by a marked change ($p = .02$ for the comparison -1 to +1 versus $p$s >.2 for all other comparisons, corrected, see Fig 2H). Importantly, the same was true in children, where we also found a significant comparison only for blocks -1 to +1, $p = .002$, but not for any other blocks, $p$s >.8. An analysis which included only participants who switched to color also revealed that younger adults implemented the new strategy with greater accuracy than children. A random effects model including factors of age group, switch group and time-period relative to switch (before versus after) revealed a significant interaction, $\chi^2(1) = 4.2$, $p = 0.039$, reflecting that the age groups did not differ before the switch, $p = .26$, but afterwards, $p = .002$ (post-hoc comparisons, adjusted).

**Response inhibition and working memory.** Finally, we investigated age differences in markers of executive control during task performance and in our covariate tasks. To characterize response inhibition, we analysed false alarm rates in *LateGo* and *NoGo* trials during the main task. This analysis showed that children and adults differed markedly in their response inhibition ability, similarly to the performance disparity seen in *standard* trials. Specifically children made significantly more premature key presses compared to younger adults (i.e., responses before the frame was displayed, henceforth "False Alarms") in *LateGo* trials (12.2% vs.1.7%, $\chi^2(1) = 11.5$, $p <.001$, Fig 3A) as well as in *NoGo* trials (11.5% vs. 1.4%, $\chi^2(1) = 13.1$, $p <.001$, Fig 3B).

Our covariate working memory and Stroop tasks also indicated significant age differences. The verbal working memory test showed that children had a lower working memory span than younger adults (6.3 vs. 10.7 correct answers, respectively, $t(30.9) = -4.3$, $p <.001$), see Fig 3C. Participants also performed a Stroop test in which they needed to respond to the ink color of a written color name (e.g., 'YELLOW' in yellow or red ink) or neutral word ('XXX', colored letters) by pressing a button. In children, RTs tended to be slower in trials with the neutral word compared to congruent trials (where color and word agreed), although not significantly, mean difference 25ms, $t(21) = 1.9$, $p = .075$. This was not true in young adults, mean difference -10ms, $t(19) = -0.92$, $p = .369$. Importantly, children had greater RT effects than adults, $t(39.6) = 2.0$, $p = .049$, Fig 3D. Note that because participants were instructed to respond to the ink color, not respond to the written word, the semantic facilitation score reflects processing of the irrelevant stimulus feature (the text of the congruent stimulus), and thus a failure of cognitive control. Surprisingly, we did not find age group differences in semantic interference (neutral—incongruent), which were -30ms and -39ms in children and younger adults, respectively, $p = .65$.

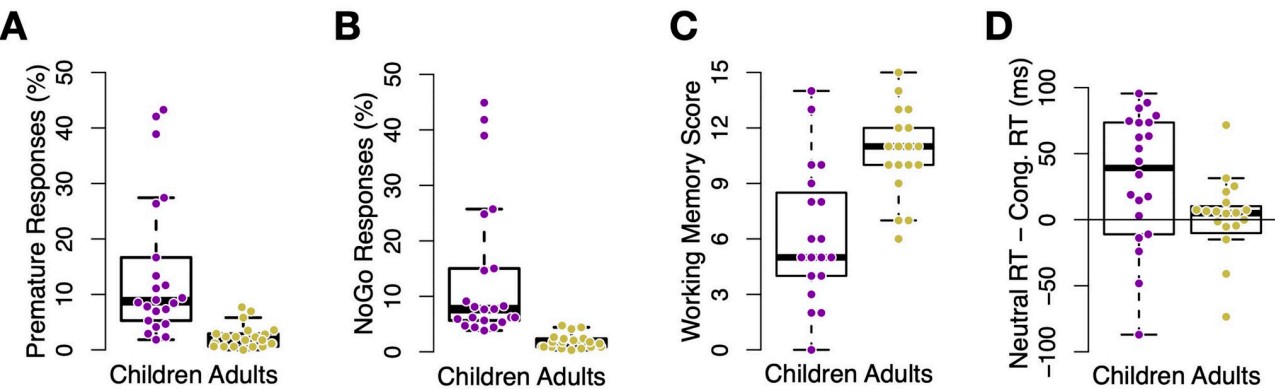

**Fig 3. Response inhibition and working memory performance.** A: Percentage of false alarms in *LateGo* trials among young adults (yellow) and children (purple) in Experiment 1, indicating significantly less errors among young adults. B: Percentage of false alarms in *NoGo*. As in panel (A), younger adults also committed less false alarms than children. C: Working memory score in a auditory digit-sorting task, reflecting the maximum number of digits that were successfully retained and ordered by each participant. Younger adult participants had on average higher working memory capacity compared to children. The reduced number of participants due to data loss caused by technical errors in the WM task. D: Average congruency effect (RT neutral—RT congruent, in ms) in the Stroop task, separately for both age groups and experiments. Younger adults showed smaller congruency effects. Each dot represents one participant, the black lines indicate boxplots. Bars represent standard error of the mean.

## Experiment 2

While Experiment 1 yielded no statistical evidence of any age differences in strategy discovery, it had a limited sample size and was characterized by large age differences in main task performance that led to differences in the number of participants who had to be excluded. In Experiment 2 we therefore repeated Experiment 1 using a slowed-down task version that allowed children to make less mistakes. Our aim was to confirm that no age differences exist even when the performance gap between adults and children is reduced and the number of exclusions equivalent. This allowed us to independently replicate our results from Experiment 1, and also offered us the possibility to address power issues by combining data across both experiments.

### Methods

**Participants.** Twenty-eight children and 21 young adults were tested in Experiment 2. Following the same exclusion criteria as in Experiment 1, three children and three adults were excluded due to being unable to perform the task during the instructed block. This resulted in an effective sample size of 25 children (10 female) with a mean age of 9.2 years (SD = 0.8, range = 8 to 10 years), and 18 young adults (4 female) with a mean age of 26.6 years (SD = 4.6, range = 20 to 35 years). All participants, and in the case of children their legal guardians, provided informed consent; the study received ethics approval identical to Experiment 1.

**Tasks.** Participants performed the same main task as in Experiment 1. To achieve better performance in children, we increased the duration of the stimulus display from 400ms to 800ms. To accommodate for the slower task, participants only received one final instructed block instead of two, reducing the block number from 10 to 9. All other aspects of the main task were identical. The working memory test was identical to Experiment 1. The Stroop task was implemented in psychoPy, but identical otherwise. Stroop data from 2 participants (1 adult) were lost due to technical error.

**Procedures.** Procedures were as in Experiment 1.

**Analyses.** Analyses followed the same principles as in Experiment 1.

## Results

**Performance on main task: Standard trials.** Main task results replicated our findings from Experiment 1. Both age groups improved performance in standard trials over the course of blocks 1-8, as reflected in a main effect of Block $\chi^2(1) = 28.3$, $p < .001$ (see Fig 4A). We also found a main effect of Age group, $\chi^2(1) = 12.9$, $p < .001$ that was driven by younger adults committing less errors than children (17.6% vs. 5.5%; Post-hoc test: $t(45.1) = 3.5$, $p < .001$). Differences in error rates between age groups persisted throughout the task, remaining present in the last two blocks before the color instruction (blocks 7-8), $t(45.1) = 3.6$, $p < .001$. No interaction between Age group and Block was found.

Likewise, RTs on standard trials in Blocks 1 to 8 differed between age groups, with a longer reaction time for children than young adults ($\chi^2(1) = 25.7$, $p < .001$) and decreased with practice, $\chi^2(1) = 21.9$, $ps < .001$ (see Fig 4B). Age group differences in RT persisted with considerable practice, as shown by planned between-group comparisons of the average RTs in blocks 7 and 8 ($t(45.1) = 4.6$, $p < .001$). After being provided with instructions to use color in Block 9, adults still outperformed children in terms of accuracy on standard trials. Error rates for adults were significantly lower than in children (2.7% vs. 6.9%, $t(43.3) = 2.2$, $p = 0.03$). In addition, children benefited more from the instructions than adults, as demonstrated by an interaction between Block (7/8 vs. 9) and age group in error rates ($\chi^2(1) = 6.5$, $p = .011$) and RTs ($\chi^2(1) = 6.2$, $p = .013$).

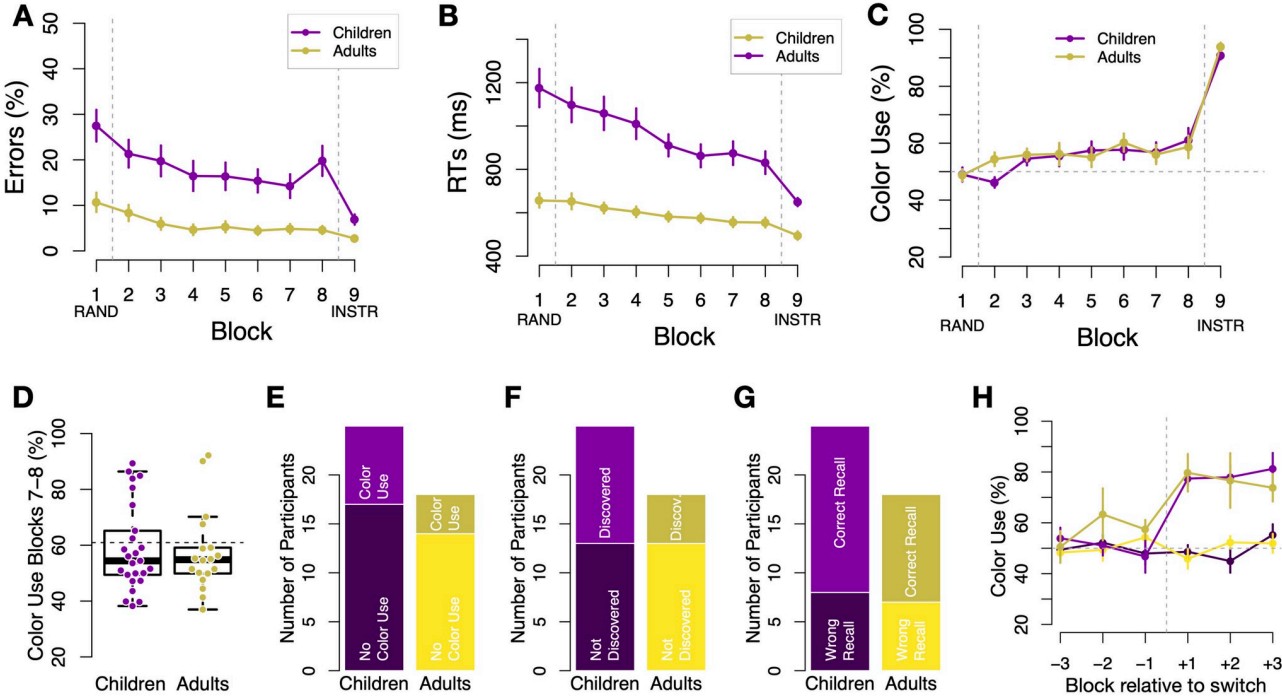

**Fig 4. Performance on standard trials and alternative strategy discovery in Experiment 2.** A: Error rates as a function of block separately for children (purple) and younger adults (yellow). Age differences in error rates persisted throughout all blocks. B: Average reaction times (RTs, in milliseconds) over blocks, also indicating sizable and persistent differences between children and young adults. Colors as in panel A. C: Percentage of color-based choices ("Color Use") in *ambiguous* trials as a function of block found in young adults (yellow) and children (purple). No significant differences were found. D: Percentage of color use in blocks 7 and 8, before instructions were given. Each dot reflects one participant. E: Proportion of participants whose behavior indicated a strategy switch towards color-based responding by blocks 7 and 8 (>60% color use). No difference was found between age groups in this measure. F: Percentage of participants self-reporting discovery of the relation between colors and corners; no age group difference was found. G: Percentage of participants able to correctly report the color-corner association after block 8, but before instructions were given. H: Percentage of color use in ambiguous trials time-locked to the mini-block in which a strategy switch was detected. Blocks on the x-axis are split in half relative to A to increase temporal resolution. Bars represent s.e.m.

**Spontaneous strategy discovery and switch.**   We next investigated participants' ability to discover and use the alternative strategy. We first examined the extent of alternative strategy use, defined identically to Experiment 1 as the proportion of color-based decisions on spatially ambiguous trials. A mixed-effects model revealed that color-based decisions increased over time (i.e., a main effect of Block, $\chi^2(1) = 16.7$, $p < .001$), but did not differ between children and adults, as in Experiment 1, see Fig 4C (i.e., no main effect of Age group, 55.6% vs 56.6% in children and young adults, respectively, $\chi^2(1) = 2.6$, $p = .11$).

Analogous to Exp. 1, we next examined color-based responding on ambiguous trials in the last two correlated blocks, before participants were given explicit instructions to use color. Again, there was no difference in the mean proportion of color-based responses between age groups (58.9% vs 57.4% in children and young adults respectively, $\chi^2(1) = 0.12$, $p = .74$, see Fig 4D). Further, applying the same binomial test criterion there were no differences between age groups in the proportion of participants who exhibited statistical evidence for above-chance color use in the last two correlated blocks ($\chi^2(1) = 0.49$, $p = .528$), with 32% (8/25) of children and 25% (4/16) of adults meeting this criteria (Fig 4E). No significant differences between groups were found when the threshold for color use was increased to 75%, $\chi^2(1) = 0.61$, $p = .675$, or decreased to 50%, $\chi^2(1) = 0.32$, $p = .752$. Moreover, neither the proportion of participants who self-reported to have discovered the color strategy (Fig 4F) nor the number of participants who accurately reported the associations between corners and colors in a questionnaire differed between age groups, ($ps > .05$, Fig 4G and 4H).

Finally, using the same method as in Experiment 1, we examined the time points when alternative color strategy was discovered. Among those participants who switched to the alternative strategy, both age groups showed the characteristic sudden onset of color use (see Fig 4H), and adults and children did not differ in when they discovered the strategy ($p = .58$). The age groups also did not differ in the extent to which they implemented the strategy before ($p = .11$) nor after ($p = 0.78$) discovering the alternative strategy (unlike in Experiment 1). In other words, there was no evidence in Experiment 2 that adults employed the new strategy with greater efficiency than children.

**Response inhibition and working memory.**   As in Experiment 1, we found age differences in cognitive control when measured in the main task as well as with separate, covariate tasks. Main task response inhibition was assessed through false alarm rates in *LateGo* and *NoGo* trials. Similar to Experiment 1, children and adults differed markedly in their response inhibition ability, paralleling the performance disparity seen in *standard* trials. Specifically, compared to young adults, children made significantly more premature key presses (i.e., responses before the frame was displayed, henceforth "False Alarms") in *LateGo* trials (5.6% vs. 0.7%, $\chi^2(1) = 8.9$, $p = .003$, Fig 5A) as well as in *NoGo* trials (6.1% vs. 0.9%, $\chi^2(1) = 11.6$, $p < .001$, Fig 5B).

Young adults also outperformed children on our additional cognitive control tasks. Children had a lower working memory span on the digit-sorting task compared to adults (4.6 vs. 8.7 correct answers, respectively, $t(33.2) = -4.9$, $p < .001$; see Fig 5C). On the Stroop task, children showed a larger facilitation effect of congruent stimuli than adults, $t(37.6) = 2.3$, $p = .029$.

## Combined analysis

### Age-differences in strategy updating versus age-differences in cognitive control and task performance

Experiments 1 and 2 yielded no evidence for differences in strategy adaptation between young adults and children, despite the fact that these groups differed substantially in task performance and cognitive control. To reduce the possibility that the lack of evidence could be

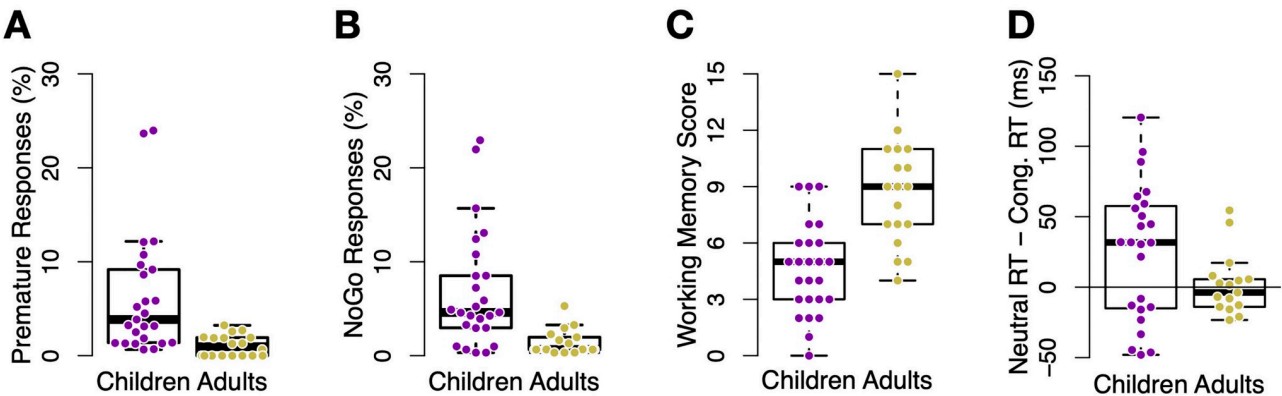

**Fig 5. Response inhibition and working memory in Experiment 2.** A: Percentage of false alarms in *LateGo* trials among young adults (yellow) and children (purple) in Experiment 2, indicating significantly less errors among young adults. B: Percentage of false alarms on *NoGo* trials. As in panel (A), younger adults also committed less false alarms than children. C: Working memory score in an auditory digit-sorting task, reflecting the maximum number of items that were successfully retained and sorted by each participant. Younger adult participants had on average higher working memory capacity compared to children. D: Average congruency effect (RT neutral—RT congruent, in ms) in the Stroop task, separately for both age groups. Children showed larger congruency effects, and more variability, than adults. Each dot represents one participant, the black lines indicate boxplots. Bars represent standard error of the mean.

related to a lack of power, we combined data across both experiments. This resulted in a sample size of 86 participants, consisting of 47 children and 39 young adults.

An analysis of this combined sample confirmed, unsurprisingly, the considerable age-differences in task performance, even at the beginning of the task (blocks 1-2): Error rates in regular trials differed between age groups (27.4% vs. 10.4%, main effect age group: $\chi^2(1) = 39.3$, $p < .001$), and the same main effect of age was found for reaction times ($\chi^2(1) = 48.4$, $p < .001$). Similarly, premature responses in LateGo and NoGo trials differed significantly across age groups early on in the task ($\chi^2(1) = 11.1$ and 18.4, both $ps < .001$) and age differences in the working memory and Stroop tasks were also confirmed in the combined sample (both $ps < .005$, see Fig 6A, left). Importantly, the same analysis did not find any evidence for differences in strategy updating. The average percentage of color use in blocks 7-8 was 58.5% in children and 60.7% in young adults, not differing significantly between age groups ($\chi^2(1) = 0.4$, $p = .51$). Among children, 27.5% of participants (13 out of 47) showed significant evidence for color use by block 7/8. Among adults, 28.2% (11/39) showed evidence for color use, which again provided no evidence that children and adults differ ($\chi^2(1) < 0.1$). We also found no evidence for differences in the switch time point: the average adult switchpoint (miniblock, only for switching adults) was 6.4, and for children it was 7.3, $t(20.6) = -0.86$, $p = .39$. Post-hoc power tests for $\chi^2$-tests revealed that with our increased sample size of 86 participants we had a power of .79 to detect effect-sizes of .3, and .45 power to detect an effect-size of .2 (analyses assuming $\alpha = .05$, two-sided).

In post experimental questionnaires, 43% of children and 39% of adults reported to have discovered the color strategy (age differences: $p = .9$). Forty-one percent of children versus 24% of adults reported to have used the strategy ($p = .16$) and 61% vs. 63% correctly reported which color as was associated with which corner ($p > .99$). Fig 6A illustrates the standardized z-scores for each age group across all measures mentioned above. Note that all measures in Fig 6 are flipped such that a positive value indicates better performance. For instance, a positive z-score for Stroop costs indicates relatively smaller Stroop costs, while a positive value for RT indicates relatively faster RTs etc.

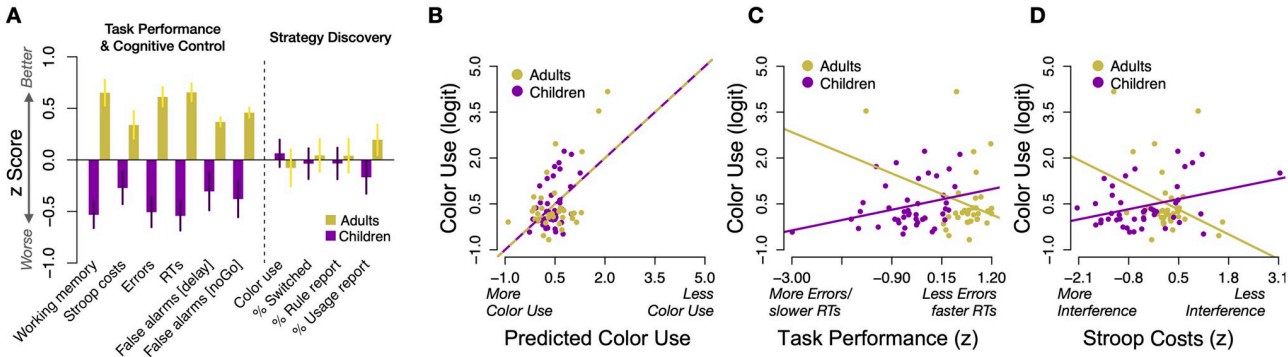

**Fig 6. Joint analyses across Experiments 1 and 2.** A: A standardized effect size (z score) was individually calculated for each performance metric for purposes of comparison. Shown metrics reflect data reported in the manuscript in Figs 2–5. Data collapsed across Experiments 1 and 2. Effects are flipped such that bars higher than 0 indicate that performance for the respective measure was better in one group, i.e. fewer errors, better working memory or faster RTs all are coded as positive values). Data from children is shown in purple, and data from adults in yellow (as in Figs 2–5). Bars represent s.e.m. B: A linear regression successfully related the logit transformed proportion of color use in Blocks 7 and 8 to the performance factors shown in panel A, $R^2 = .27$. C: The regression model indicated several interactions between age group and cognitive performance. The scatterplot shows task performance (better performance from left to right) is related to logit transformed color use. D: The scatterplot shows that Stroop semantic facilitation effect (better performance, i.e. *less* RT costs, from left to right) also had reversed association with color use in young adults versus children. Each dot represents one participant. Lines reflect regression slopes of simple models including only the illustrated factors.

The sizable differences in task performance and cognitive control therefore stand in contrast to the lack of differences in strategy adaptation. To formally test this impression, we performed a linear mixed effects model in which the z-scored performance in each measure was treated as the dependent variable, and factors Agegroup and Cognitive Ability (cognitive control/task performance versus strategy updating, see Fig 6A) were tested. As expected, this model indicated a clear interaction of Agegroup and the type of cognitive ability, $\chi^2(1) = 15.6$, $p < .001$.

We also tested whether the lack of age differences in the proportion of color use in ambiguous trials could be explained by low reliability in our measures of strategy discovery. The split half correlation between color use in odd (2, 4, 6, 8) and even (3, 5, 7) blocks was $r = .84$ (despite the fact that color use changed across time for some people, as we show in this paper). Constraining the analysis only to periods after the strategy was implemented yielded a correlation of $r = .86$. Hence, our measures of strategy adaptation appear highly reliable.

## Relations between strategy discovery, task performance, working memory, and Stroop performance

Finally, we investigated whether measures of cognitive control and task performance were related to the use of the alternative strategy. We used a linear regression model to predict the logit transformed proportion of color use in ambiguous trials in blocks 7 and 8, using the indicators of cognitive functioning discussed above. Because the measures for standard trial performance (RTs and errors) and response inhibition (premature responses in LateGo and NoGo trials) were highly colinear ($r = .35$ and $r = .89$, respectively), we z-scored and then averaged the affected pairs of variables into singular factors (e.g. on-task performance: z-scored RT + z-scored Error rate). Hence the model included five factors in total: age group, on-task performance, on-task response inhibition, Stroop semantic facilitation effect, and working memory capacity. All main effects were included as well as all pairwise interactions between age group and each of the performance measures. A baseline model that included only age group as a predictor did not indicate any main effect of age group ($p = .73$) and had significantly

worse fit than the full model ($r^2 = .27$ vs. $r^2 = .001$, Akaike Information Criterion, AIC: 198 vs 207, $p < .001$), see Fig 6B. A stepwise model selection procedure based on AIC confirmed that only the WM main effect and the WM interaction with age group could be removed from the model without loss of fit (final model AIC: 196). Importantly, the full model indicated significant interaction effects of age group and task performance ($p = .02$) as well as age group and stroop semantic facilitation ($p < .001$) on the amount of color use. These interactions reflected that the relationship between strategy updating and the other cognitive abilities was negative among young adults, but positive among children. Specifically, better task performance (less errors/faster RTs) was associated with less strategy adaptation in young adults, but the reverse was true among children (Fig 6C, $r = −.29$ vs. $r = .30$, simple correlations are reported to illustrate the effect). Similarly, less Stroop RT costs were associated with less color use among young adults, $r = −.32$, while the opposite was true among children, $r = .35$, see Fig 6D. Note, however, that only the effect involving Stroop performance (6D), but not the effect of task performance (6C), survived a reanalysis with a robust regression framework.

## Discussion

In two experiments we compared children's and adults' ability to discover possible strategy improvements during task execution. A strategy adaptation occurred when participants changed how they selected their responses throughout the task although a viable response rule was provided at the beginning of the experiment. The instructed task rule allowed error-free task execution, no error feedback was given, and the possibility that an alternative strategy could be found was not mentioned by the experimenters. Strategy improvements were therefore a product of participant's self driven exploration of alternative stimulus-response rules.

Our results showed that strategy adaptations occurred equally often in children and adults. This finding contrasted with the superior performance of adults in all other cognitive abilities that were measured in the same sample, in particular in task execution, working memory, and cognitive control abilities. Flexible strategy updating therefore presents a remarkable exception to the well documented protracted development of decision-making relevant functions in children such as cognitive control [7, 21], rule following [4], model based decision making [29] and choice exploration strategies [37].

Notably, we found different associations between the amount of color use (strategy adaptation) and performance in other tasks among children versus adults. Most interestingly, the Stroop (semantic facilitation) effect correlated positively with color use among children ($r = .35$, whereby larger Stroop effects indicate less interference, see Fig 6D). This hints at a 'beneficial' effect of more cognitive control. Note, however, that we found no age differences in semantic facilitation in our Stroop test, the reasons for which remain unclear and could suggest unreliable measurement. In young adults, in contrast, we found a *negative* relation between the Stroop effect and color use, $r = −.32$, indicating that young adults with better executive functions were less influenced by the color, see Fig 6D. The same pattern of results was found when investigating task performance.

While the present study was not designed to specifically examine the question about the factors influencing strategy discovery, the available data thus could hint at two possible explanations. On the one hand, relatively better task and Stroop performance could reflect different mechanisms in adults versus children. Whereas in adults good performance mainly reflects attentional focus that is detrimental to strategy updating, in children relatively good task performance could reflect different factors, such as better encoding of the instructions or better motor control.

Interestingly, selective attention is thought to be mature relative to inhibitory control among children in the present study; 8-10 year olds have been found to show only small

deficits on selective attention tasks, but large differences on response inhibition tasks, relative to adults [38]. This protracted development of response inhibition is congruent with the marked differences in task performance (e.g., on LateGo and No-Go trials) in our current study, and suggests that it may be response inhibition, rather than attentional focus more broadly, that contributes to the ability to spontaneously discover alternative solutions.

On the other hand, given the overall performance difference between children and adults it also seems possible that the opposite-sign relations to strategy switching reflect an inverted U-shaped relationship between attentional mechanisms and strategy switching. Further investigations are therefore needed that shed light on the factors that facilitate and impede strategy discovery, for instance using Bayesian models to computationally capture parameters of exploitation and exploration directly. Furthermore, more differentiated measures of selective attention, response inhibition, and update focused working memory measures such as n-back or AX-CPT tasks should be utilized to tease apart the specific aspects of cognitive control that relate to strategy switching in children and adults, respectively.

In addition, given the between subject nature of the effects, larger sample sizes that yield higher power for detecting small difference between age groups will be needed. In addition, note that any novel insights obtained from combining Experiments 1 and 2 ignore the fact that doubling the stimulus display might have affected the deployment of the color strategy. Hence, new experiments using a constant stimulus display with a larger sample size are warranted.

Which computational properties allow a decision-maker to find new solutions within previously ignored environmental structure remains an overall unsolved problem that relates to the general question of representation learning mechanisms in the brain [39–41], a topic that has also been considered in developmental psychology, e.g. [42]. It also remains unclear how the high levels of flexible updating could be neurally implemented in the still developing brain. Our own investigation in younger adults suggested that the spontaneous change in strategy relied on a internal simulation mechanism in medial prefrontal cortex (mPFC). In children, mPFC displays a complex structural maturation trajectory that differs between its subregions [43], with the orbital parts following an early maturation pattern, whereas the dorsal parts follow a late maturation pattern. The cluster of mPFC found in [32] corresponds to the region that goes through structural transition between 8 and 10 years of age. This implies that our sample of children could be characterized by substantial inter-individual differences in mPFC structure. Direct measurement of mPFC structure could therefore be an interesting subject of investigation.

In addition, it remains unclear whether children's brains exhibit similar dynamics in long range brain activity correlations that have been associated with the task used here [36], given the marked changes in brain network segregation observed in children [3]. This may be relevant insofar as prefrontal network dynamics have been linked to the balance between cognitive stability and flexibility [44], suggesting that the stable states that correspond to task sets representations can be thought of as basins in a potential landscape of network state. According to this view, deeper basins are related to cognitive stability and efficient task execution, while shallower basins imply less effort to switch but higher susceptibility to distraction. In line with this idea it has been found that depth of the attractor state, as indexed by functional coupling between prefrontal areas, is related to how readily individuals switch from one task state to another in the light of ambiguous task cues [45]. Therefore, the development of attractor stability of prefrontal networks may be a useful topic for future investigations, see also [3].

Several further shortcomings of the study need to be acknowledged. First, due to limitation in testing time of children, we did not include a comprehensive battery of cognitive control. One may speculate that updating, another aspect of executive function [46], could be relevant for strategy adaptation and should be included in future studies. Second, developmental

constraints to strategy updating may stem from different reasons, namely difficulties in discovering an alternative strategy vs. switching to another strategy. Our questionnaire results suggest that, numerically, a higher percent of children (41%) than adults (24%) reported to have used the strategy. While this difference was not significant and based on subjective reporting, the lack of age difference in actual color strategy use could hint at difficulty in switching even if the alternative strategy was discovered. In relation to this, the difficulty in switching could be related to contextual factors such as compliance to task instruction, which may differ for participants of different age group and will be of interest for future studies.

In summary, the present study has shown that the ability to perform strategy adaptations presents a remarkable exception from children's comparatively limited decision making skills, such as executing simple task rules, holding information in working memory and inhibiting prepotent responses. The comparatively well developed ability to discover novel strategies for a known task in children might offer a unique opportunity for educators in fostering learning in children. More generally, our findings highlight that the development of cognitive functions in children might result in complex dynamics of abilities that rely on the interaction of several cognitive functions.

## Acknowledgments

We thank Azzurra Ruggeri for comments on our manuscript.

## Author Contributions

**Conceptualization:** Dorit Wenke, Robert Gaschler, Yee Lee Shing.

**Formal analysis:** Nicolas W. Schuck, Amy X. Li, Anika T. Loewe.

**Funding acquisition:** Nicolas W. Schuck, Yee Lee Shing.

**Investigation:** Amy X. Li, Destina S. Ay-Bryson, Anika T. Loewe.

**Software:** Nicolas W. Schuck.

**Supervision:** Dorit Wenke, Robert Gaschler, Yee Lee Shing.

**Visualization:** Nicolas W. Schuck.

**Writing – original draft:** Nicolas W. Schuck, Amy X. Li.

**Writing – review & editing:** Nicolas W. Schuck, Amy X. Li, Dorit Wenke, Destina S. Ay-Bryson, Anika T. Loewe, Robert Gaschler, Yee Lee Shing.

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
