## [Decision Letter · Decision Letter 0]

22 Dec 2021

PONE-D-21-37011Spontaneous discovery of novel task solutions in childrenPLOS ONE

Dear Dr. Schuck,

Thank you for submitting your manuscript to PLOS ONE. After careful consideration, we feel that it has merit but does not meet PLOS ONE’s publication criteria as it currently stands. As you can see from the reviews, both experts find this study interesting in principle. However, they raise a number of important points concerning analyses, presentation of the results and theoretical interpretation. These points need to be fully addressed before the manuscript can be considered suitable for publication, and we therefore invite you to submit a revised version of the manuscript.

We look forward to receiving your revised manuscript.

Kind regards,

Maria Wimber

Academic Editor

PLOS ONE

Journal Requirements:

"We thank Azzurra Ruggeri for comments on our manuscript. NWS was funded by an Independent Max Planck Research Group grant awarded by the Max Planck Society and a Starting Grant from the European Union (ERC-2019-StG-REPLAY-852669). DW was funded by DFG grant WE2852/3-1. YLS was funded by a Minerva Research Group by the Max Planck Society, a Starting Grant from the European Union (ERC-2018-StG-PIVOTAL 758898), and a Fellowship from the Jacobs Foundation (JRF 2018-2020). AL is supported by the International Max Planck Research School on Computational Methods in Psychiatry and Ageing Research (IMPRS COMP2PSYCH)."

"NWS was funded by an Independent Max Planck Research Group grant awarded by the Max Planck Society (www.mpg.de) and a Starting Grant from the European Union (ERC-2019-StG-REPLAY-852669, www.erc.europa.eu/). DW was funded by DFG grant WE2852/3-1 (www.dfg.de). YLS was funded by a Minerva Research Group by the Max Planck Society (www.mpg.de), a Starting Grant from the European Union (ERC-2018-StG-PIVOTAL-758898, www.erc.europa.eu), and a Fellowship from the Jacobs Foundation (JRF 2018-2020, www.jacobsfoundation.org). AL is supported by the International Max Planck Research School on Computational Methods in Psychiatry and Ageing Research (IMPRS COMP2PSYCH, www.mps-ucl-centre.mpg.de). 

Additional Editor Comments:

Both expert reviewers find this study interesting, however, they raise a number of important points that need to be addressed before the manuscript can be considered suitable for publication.

Reviewers' comments:

Reviewer's Responses to Questions

**Comments to the Author**

1. Is the manuscript technically sound, and do the data support the conclusions?

Reviewer #1: Yes

Reviewer #2: Partly

2. Has the statistical analysis been performed appropriately and rigorously? 

Reviewer #1: Yes

Reviewer #2: Yes

3. Have the authors made all data underlying the findings in their manuscript fully available?

Reviewer #1: No

Reviewer #2: No

4. Is the manuscript presented in an intelligible fashion and written in standard English?

Reviewer #1: Yes

Reviewer #2: Yes

5. Review Comments to the Author

Reviewer #1: This manuscript presents two studies and a combined analysis of both samples to analyze the strategy adaptation of children in a visual choice task. The data presented suggested that children and adults have the same amount of strategy shift in the task, although the performance of children is lower than adults. The manuscript is well written, and its statistical method is very clear and elegant. Overall, this manuscript is almost publication-ready, but small changes could benefit the paper. Thus, I recommend this paper be accepted after minor revision.

This review is signed,

-Paulo Laurence

Major points:

1) The authors mentioned that their lack of evidence of their results in studies 1 and 2 could be an effect of small power (page 24, lines 433-434). Their solution was to combine the data from studies 1 and 2. I think that authors could also conduct a post-hoc achieved power analysis in order to statistically present what is the power that they found when combining both samples.

Minor points

1) The plots are presented in shades of blue and red. Thinking that colorblindness affects ~4.5% of the population, it can be interesting to change the color of the plots to shades that are more colorblind-friendly. One example is the Viridis package of R, which presents dark purple (#440154FF) and light yellow (#FDE725FF) as opposing colors. Other examples of colorblind-friendly colors are #999999, #E69F00, #56B4E9, #009E73, #F0E442, #0072B2, #D55E00, #CC79A7. This change can help the interpretability of the paper. I do not think this is a must-do change to the paper, but it would benefit it.

2) Plot A and B of Figure 5 is missing the x-axis title. Please, add the x-axis title in these plots

Reviewer #2: The authors address a worthwhile question — does children’s more diffuse attention aid them in the discovery of novel strategies? The authors present a study comparing children’s ability to discover novel strategies with that of adults and examined how this related to cognitive control abilities. They found that children and adults did not differ in their discovery of novel strategies despite differences in their overall performance in the task, working memory capacity, and response inhibition.

1. I question whether it’s valid to combine the datasets from experiment 1 and 2. Doubling the stimulus display time is a considerable change and seems like it could have affected the deployment of the instructed color strategy (i.e. in experiment 2, there is not a difference in color use between children and adults after instruction).

2. The following points concern the discussion.

a. On page 29, the results are discussed in light of bayesian accounts of developmental change (e.g. Children have weaker priors and hence rely more on the data they encounter). The authors point out that the instructed spatial strategy implements a prior belief that should be shared between children and adults. But, the discussion is very vague regarding why children would be regularizing more towards this prior belief than adults which is what their data from the first experiment suggests.

b. On page 30, mPFC and its role in novel strategy discovery and its ongoing development at 8-10 is discussed. However, it is not fully addressed what it means that mPFC is still under development yet these behavioral results demonstrate that children are able to discover novel strategies at the same rate as adults.

c.How do these results relate to differences in the developmental trajectories of selective attention and response inhibition (Booth, 2003 Neuroimage; Tipper, 1989, Journal of Experimental Child Psychology)? Particularly because of the child age range examined. 8-10 year olds are thought to have relatively mature selective attention but response inhibition continues to develop into adolescence.

3. The following points concern the Stroop results

a. On page 18, there is a footnote explaining that semantic facilitation reflects a failure of cognitive control and reports no age difference in semantic interference. I suggest moving this out of the footnote and providing further explanation for why semantic facilitation reflects a failure of cognitive control. Is this because it demonstrates that they are attending to an irrelevant feature (text)? This feels important for motivating the correlations between main task performance and semantic facilitation later on.

b. There is a difference between age groups for the semantic facilitation effect but not the semantic interference effect. While not critical to the argument of the paper, I think a quick interpretation of this asymmetry in the discussion would be beneficial in understanding what sorts of inhibitory control show age related differences in this study.

c. What is “Stroop Costs” in Figure 6a? Is it the same measure referred to as “Stroop effect” in Figure 6D?

4. On pages 10-11 lines 190-193, it states that 7 participants’ data was lost due to human and technical errors and then 2 participants’ data was lost to technical errors. Does this mean 9 participants’ data was lost?

5. Are the reported correlations on page 26 line 493-495 Pearson? From Figure 6c, it looks like the adult correlation may be driven by the one participant with a very high error rate and very high color use. A spearman rank correlation would be less sensitive to this outlier.

6. There are some places where the p-value is reported by not the test statistic (e.g. page 20, line 380 and 382).

7.The following are analyses that I don’t think are critical for the claims of the paper but I think could clarify some of the results.

a. Were there any differences in the distributions of estimated switch points between child and adult color users?

b. When child and adult color users (prior to the instructed block) accidentally respond on NoGo Trials, do they tend to respond with the option that corresponds with the color or randomly? Are there age differences in this?

6. PLOS authors have the option to publish the peer review history of their article (what does this mean?). If published, this will include your full peer review and any attached files.

Reviewer #1: **Yes: **Paulo Guirro Laurence

Reviewer #2: No

---

## [Author Response · Author response to Decision Letter 0]

1 Mar 2022

See attached "Response to Reviewers.pdf" Document

---

## [Decision Letter · Decision Letter 1]

17 Mar 2022

Spontaneous discovery of novel task solutions in children

PONE-D-21-37011R1

Dear Dr. Schuck,

We’re pleased to inform you that your manuscript has been judged scientifically suitable for publication and will be formally accepted for publication once it meets all outstanding technical requirements.

Kind regards,

Maria Wimber

Academic Editor

PLOS ONE

Additional Editor Comments (optional):

Please do make an effort to share data & code for this publication in line with journal policy.

Reviewers' comments:

Reviewer's Responses to Questions

**Comments to the Author**

1. If the authors have adequately addressed your comments raised in a previous round of review and you feel that this manuscript is now acceptable for publication, you may indicate that here to bypass the “Comments to the Author” section, enter your conflict of interest statement in the “Confidential to Editor” section, and submit your "Accept" recommendation.

Reviewer #1: All comments have been addressed

Reviewer #2: All comments have been addressed

2. Is the manuscript technically sound, and do the data support the conclusions?

Reviewer #1: Yes

Reviewer #2: Yes

3. Has the statistical analysis been performed appropriately and rigorously? 

Reviewer #1: Yes

Reviewer #2: Yes

4. Have the authors made all data underlying the findings in their manuscript fully available?

Reviewer #1: No

Reviewer #2: Yes

5. Is the manuscript presented in an intelligible fashion and written in standard English?

Reviewer #1: Yes

Reviewer #2: Yes

6. Review Comments to the Author

Reviewer #1: Dear Authors,

I find you revision adequate addressing my comments. Reviewer #2 raised a significant point regarding the outlier in figure 6C, but the authors’ answer and changes in the text were satisfactory. Thus, I think this manuscript is ready for publication. Thank you for this interesting paper.

This review is signed,

-Paulo Laurence

Reviewer #2: The authors have addressed all my concerns. One small thing —The manuscript currently does not include links to repositories containing data and analysis code. I'm assuming these will be inserted later.

7. PLOS authors have the option to publish the peer review history of their article (what does this mean?). If published, this will include your full peer review and any attached files.

Reviewer #1: **Yes: **Paulo G. Laurence

Reviewer #2: No

---

## [Editor Report · Acceptance letter]

16 May 2022

PONE-D-21-37011R1 

Spontaneous discovery of novel task solutions in children 

Dear Dr. Schuck:

I'm pleased to inform you that your manuscript has been deemed suitable for publication in PLOS ONE. Congratulations! Your manuscript is now with our production department. 

Kind regards, 

on behalf of

Prof. Maria Wimber 

Academic Editor

PLOS ONE